# Quality of Surgical Outcome Reporting in Randomised Clinical Trials of Multimodal Rectal Cancer Treatment: A Systematic Review

**DOI:** 10.3390/cancers16010026

**Published:** 2023-12-20

**Authors:** Joanna Janczak, Kristjan Ukegjini, Stephan Bischofberger, Matthias Turina, Philip C. Müller, Thomas Steffen

**Affiliations:** 1Clinic for General and Visceral Surgery, Hospital for the Region Fürstenland Toggenburg, CH-9500 Wil, Switzerland; joanna.janczak@srft.ch; 2Department of Surgery, Hospital of the Canton of St. Gallen, CH-9007 St. Gallen, Switzerland; kristjan.ukegjini@kssg.ch (K.U.); stephan.bischofberger@kssg.ch (S.B.); 3Department of Surgery and Transplantation, University Hospital Zurich, CH-8091 Zurich, Switzerland; matthias.turina@usz.ch; 4Department of Surgery, Clarunis—University Centre for Gastrointestinal and Hepatopancreatobiliary Diseases, CH-4002 Basel, Switzerland; philip.mueller@clarunis.ch

**Keywords:** rectal cancer, radiotherapy, chemotherapy, randomised controlled trials, surgical quality

## Abstract

**Simple Summary:**

Reporting surgical outcome and complication data in RCTs on rectal cancer is important because it is the basis for judging whether the results of a study warrant a change in clinical practice. In this review, we systematically analysed the quality of reporting in RCTs and the completeness rate of reporting of surgical outcomes and complication data. We found that only 2% (N = 7) of the RCTs met all 14 reporting criteria, and nearly half (N = 168, 49%) completed the procedure-specific quality criteria noted in the article. The most underreported criteria included complication severity (15% of articles).

**Abstract:**

Introduction: Randomised controlled trials (RCTs) continue to provide the best evidence for treatment options, but the quality of reporting in RCTs and the completeness rate of reporting of surgical outcomes and complication data vary widely. The aim of this study was to measure the quality of reporting of the surgical outcome and complication data in RCTs of rectal cancer treatment and whether this quality has changed over time. Methods: Eligible articles with the keywords (“rectal cancer” OR “rectal carcinoma”) AND (“radiation” OR “radiotherapy”) that were RCTs and published in the English, German, Polish, or Italian language were identified by reviewing all abstracts published from 1982 through 2022. Two authors independently screened and analysed all studies. The quality of the surgical outcome and complication data was assessed based on fourteen criteria, and the quality of RCTs was evaluated based on a modified Jadad scale. The primary outcome was the quality of reporting in RCTs and the completeness rate of reporting of surgical results and complication data. Results: A total of 340 articles reporting multimodal therapy outcomes for 143,576 rectal cancer patients were analysed. A total of 7 articles (2%) met all 14 reporting criteria, 13 met 13 criteria, 27 met from 11 to 12 criteria, 36 met from 9 to 10 criteria, 76 met from 7 to 8 criteria, and most articles met fewer than 7 criteria (mean 5.5 criteria). Commonly underreported criteria included complication severity (15% of articles), macroscopic integrity of mesorectal excision (17% of articles), length of stay (18% of articles), number of lymph nodes (21% of articles), distance between the tumour and circumferential resection margin (CRM) (26% of articles), surgical radicality according to the site of the primary tumour (R0 vs. R1 + R2) (29% of articles), and CRM status (38% of articles). Conclusion: Inconsistent surgical outcome and complication data reporting in multimodal rectal cancer treatment RCTs is standard. Standardised reporting of clinical and oncological outcomes should be established to facilitate comparing studies and results of related research topics.

## 1. Introduction

Only little is known about the quality of reporting of surgical outcomes and complication data in randomised controlled trials (RCTs) on the multimodal treatment regimens for rectal cancer. The results of RCTs may have a substantial impact on future treatment regimens. In multimodal treatment regimens, the quality of surgical treatments remains of the utmost prognostic relevance to patients with rectal cancer. Over the last 40 years, rectal cancer treatment has made impressive progress by developing and implementing differentiated quality criteria for surgical treatment, including multimodal therapy concepts [1]. Heald et al. [2] proposed the notion of total mesorectal excision (TME). This has become the surgical standard for cancer of the middle and lower third of the rectum [2,3,4]. A microscopically negative circumferential resection margin (CRM), sufficient lymph node retrieval, and the integrity of the mesorectal fascia are all associated with lower local and distant recurrence rates and better long-term survival [5,6].

In studies, RCTs lower the risk of selection bias. However, this study design is not necessarily associated with high practical relevance. Reporting surgical outcomes and complication data in RCTs is essential because it is the basis for judging whether the results of a study warrant a change in clinical practice. On the contrary, low-quality data reporting or omitting critical data relevant to the overall outcome may result in misleading reports from studies. Different scales and checklists for assessing the quality of reporting in RCTs have been developed [7,8,9], including the Jadad scale, which was developed using standard techniques of scale development [10].

Another critical aspect of evaluating the efficacy of rectal cancer treatment is the need for standardised reporting of surgical outcome and complication data due to the mostly elderly population of rectal cancer patients and their associated medical comorbidities. In addition, the success of multimodal therapies for treating rectal cancer often relies on the timing of chemotherapy or radiotherapy with surgical intervention, and whether these therapies are considered successful can be affected by surgical outcome criteria and perioperative complications. Perioperative complication grading systems [11,12] and, most recently, standard criteria for reporting surgical outcomes have been established [13,14,15].

We hypothesise that the quality of reporting of surgical outcome and complication data in RCTs is high. To address this hypothesis, we conducted this systematic review. This study aimed to measure the quality of the reporting of surgical outcome and complication data in RCTs of rectal cancer treatment and to analyse the change in the reporting quality over 40 years.

## 2. Materials and Methods

This systematic review complied with the PRISMA guidelines [16]. This review was prospectively registered in an international prospective registry for systematic reviews under the study number review registry 1682.

### 2.1. Literature Search

To identify studies, we performed a systematic literature search of EMBASE, MEDLINE (via PubMed), and Cochrane Central Register of Controlled Trials using the following strings: ((“rectal cancer” OR “rectal carcinoma”) AND (radiation OR radiotherapy)) (Figure 1). The results were limited to from January 1982 to 10 February 2022. Two of the authors (JJ and KU) independently checked whether the full-text articles were eligible. Disagreements were clarified through a one-by-one discussion.

The time interval from 1982 to 2022 was chosen because RCTs were infrequently published in the journals studied before this period and because of the ground-breaking insights in rectal cancer treatment Heald promoted at that time [2,17]. A subgroup analysis was performed to compare publications published before and after the year 2000, as the implementation of the TME concept and related procedure-specific criteria occurred primarily on a global scale in the new millennium. The year 2001 was chosen as the cut-off point because this was the time when these procedures were being performed—specific criteria were already in place or were recommended in many guidelines.

### 2.2. Inclusion and Exclusion Criteria

Articles that met the following criteria were selected for inclusion:(1)population—patients with rectal cancer;(2)treatment—radiotherapy as part of the regimen;(3)outcomes—treatment methods;(4)study design—RCTs;(5)language of search results—limited to the English, German, Polish, or Italian languages.

The excluded articles were non-RCTs, literature reviews, observational epidemiological studies (cohort and case-control designs), case series, case reports, articles describing other forms of rectal cancer treatment without radiotherapy in the treatment regimen, and studies including cancers other than rectal cancer and articles are written in a language different than the above mentioned. All duplicate articles were removed, and database limits were utilised to exclude paediatric and anatomical/cadaver articles.

### 2.3. Outcome Parameters

The primary outcome was the quality of reporting in RCTs and the completeness rate of reporting of surgical outcome and complication data; this outcome was assessed based on fourteen criteria (Table 1).

The following secondary outcomes were also collected and assessed: study period, study design, the population size of rectal cancer patients, clinical data, type of treatment, country of origin, publication year, the absolute number of citations, single or multicentre study design, sample size, study intervention and control groups, study endpoints, the conclusion of the study, Quirke grade as a marker of the quality of surgical resection, the field of medical speciality, Scientific SCImago Journal Ranking, journal impact factor (IF) according to the Web of Science Group, and the type of surgical procedures performed.

An additional outcome measure was to objectively evaluate changes in procedure-specific quality criteria over time, particularly with respect to their initial consultation in previous literature.

### 2.4. Quality of Reporting

We modified previously published criteria related to the completeness of surgical outcomes and complication reporting to rate the articles [13]. Fourteen critical components were evaluated (Table 1). Providing information on surgical interventions using the 14 components allows a complete understanding of the quality of outcomes and complication reporting. As initially, the 14 criteria did not address oncological radicality in rectal cancer surgery, this was assessed separately. In particular, the category “Procedure-specific quality” was adjusted, as this outcome is highly relevant in rectal cancer surgery because they are all associated with lower local and distant recurrence rates and better long-term survival. If the items described were mentioned in the assessed articles, 1 point was awarded for each item. If an item was not noted, no point was given.

In turn, the category “quality of reporting in RCTs” was created, as these outcomes are often of interest in RCTs. The Jadad scale [10], shown previously as reliable and valid, was used. The Jadad scale is a 3-item scale covering the randomisation method, the blinding method, and withdrawals/dropouts (Appendix A). For each item, 1 point was assigned to a study if it was described as randomised or double-blinded or had discontinuations/failures. If the described randomisation or blinding method was judged appropriate, 1 point was assigned for that item.

In contrast, no point was awarded for this item if the described randomisation or blinding method was deemed inadequate. In the original Jadad scale, only double blinding was considered. The scale was modified to include single blinding because double blinding is not always possible in surgical procedures. The blinding method was considered appropriate if the item indicated who was involved in blinding and, depending on the type of intervention, possible additional measures to ensure blinding. The final quality score for each article ranged from 0 (lowest quality) to 5 points (highest quality).

We searched (in August 2023) the official relevant websites for information regarding the main characteristics of the journal IF, according to the SCI Journal website, [18] and the SJR indicator, provided by the SCImago journal and country rank. [19] Journals were classified by the IF, with an IF of 10 or more considered excellent, a value between 3 and 10 considered very good, a value between 1 and 3 considered good, and a value below 1 considered moderately good.

### 2.5. Statistical Analysis

Summary statistics were tabulated via established norms. Baseline characteristics were summarized using counts and percentages. Weighted overall rates were calculated for dichotomous data. Means were determined using normally distributed continuous data. Microsoft Excel was used to analyse the data. A *p*-value less than 0.05 (2-sided) was considered to be statistically significant.

Median and range scores for the quality were calculated for all articles and each journal. Student’s *t*-test was used to compare sample distributions. Differences were statistically significant if the *p*-value was less than 0.05.

### 2.6. Ethical Statement

The Ethical Committee of the Cantonal Hospital St. Gallen reviewed this study. Because all data used are publicly available, the study was exempt from further oversight and requirements concerning informed consent. The Strengthening the Reporting of Observational Studies in Epidemiology (STROBE) reporting guidelines were applied [20].

## 3. Results

We initially identified 529 articles (Figure 1). After excluding 27 duplicates, titles and abstracts were assessed based on the inclusion criteria. A full-text review was performed on all selected articles, after which 340 were further analysed.

### 3.1. Description of Included Studies

The 340 articles identified included outcomes for multimodal therapy in 143,531 rectal cancer patients (Appendix A). A total of 77.1% (N = 262) of the articles had 100 or more participants. Overall, 28.9% (N = 98) of these articles recruited 500 or more participants, and the median number of participants per study was 141.5 (IQR, 106.2–582.2). Most papers comprised multisite RCTs (N = 242, 71.2%); 98 (28.8%) RCTs were performed at single sites. Most RCTs were initiated at a European research site (N = 243, 71.3%), and 335 (98.5%) were open-label. Seventy-three articles (21.5%) focused on overall survival, 73 (21.4%) focused on locoregional recurrence, 51 (15%) focused on disease-free survival, 51 (15%) concentrated on pathological complete response, and 19 (5.6%) focused on postoperative morbidity and mortality. Of the 340 articles reviewed, 69 (20.3%) were published in surgical journals, 23 (6.8%) in medical journals, 135 (39.7%) in oncology journals, and 69 (20.3%) in radiology journals. Most studies (N = 295, 86.7%) were published in journals with an IF of 3 or higher. Nearly 30% (N = 101) were published in high-impact journals with an IF of 10 or higher, such as The New England Journal of Medicine (N = 7, 2.1%) [21,22,23,24,25,26,27], The Lancet (N = 3, 0.9%) [28,29,30], Journal of Clinical Oncology (N = 36, 10.6%) [31,32,33,34,35,36,37,38,39,40,41,42,43,44,45,46,47,48,49,50,51,52,53,54,55,56,57,58,59,60,61,62,63,64,65,66], The Lancet Oncology (N = 11, 3.2%) [67,68,69,70,71,72,73,74,75,76,77], Annals of Oncology (N = 17, 5%) [78,79,80,81,82,83,84,85,86,87,88,89,90,91,92,93,94], Journal of the American Medical Association Oncology (N = 3, 0.9%) [95,96,97], Journal of the American Medical Association Surgery (N = 2, 0.6%) [98,99], Journal of the National Cancer Institute (N = 5, 1.5%) [100,101,102,103,104], Annals of Surgery (N = 12, 3.5%) [105,106,107,108,109,110,111,112,113,114,115,116], Clinical Cancer Research (N = 4, 1.2%) [117,118,119,120], and Cancer Communications (N = 1, 0.3%) [121].

### 3.2. Quality of Reporting in RCTs

All 340 articles described a randomisation method (Table 2). Five articles reported double or single blinding, and 335 reported a correctly performed randomisation method and assessment. The description of withdrawals and dropouts were reported in all 340 articles. The median scale score for the 340 articles was 4 points (range, 4–5).

### 3.3. Critical Appraisal

Of the 14 criteria related to the completeness of reporting of surgical outcomes and complication data, only seven articles (2%) [61,70,114,122,123,124,125] met all the criteria (Figure 2). Four articles (<1.0%) [71,126,127,128] met 13 criteria, and 27 articles (8%) met either 11 or 12 criteria. Additionally, 36 articles (11%) met from 9 to 10 criteria, 76 (22%) met from 7 to 8 criteria, 68 (20%) met from 5 to 6 criteria, 79 (23%) met from 3 to 4 criteria, and 43 (13%) met from 1 to 2 criteria. A mean of 6.1 ± 3.1 (SD) criteria and a median of 5.5 (IQR 5) were met. There was a linear trend in the quality of the articles, as reflected by the mean criteria met, over time (Appendix A).

### 3.4. Quality of Reporting of Surgical Outcome and Complication Data

Table 3 shows specific reporting criteria and compliance rates for each primary endpoint. The method of data accrual (N = 340, 100%) and the duration of follow-up (N = 340, 100%) were reported consistently. Commonly underreported criteria included the complication severity (N = 51, 15%), macroscopic integrity of mesorectal excision (N = 57, 17%), length of stay (N = 61, 18%), number of retrieved lymph nodes (N = 70, 21%), distance between the tumour and CRM (N = 90, 26%), surgical radicality according to the site of the primary tumour (N = 100, 29%), and CRM status (N = 128, 38%). In the articles reporting complication severity, 4 articles (1.2%) used the simple classification of major versus minor complications, and 47 articles (13.8%) used validated grading systems [11,12]. Of the four articles using the major versus minor categorisation, four definitions were used to describe what constituted a major complication. Procedure-specific quality was reported in 172 articles (51%) (mean criteria = 1.3 ± 1.6 (SD); median criteria = 0 (range 0–5)); however, only 20 articles (5.9%) [61,69,70,71,73,74,89,98,114,115,122,123,124,125,126,129,130,131,132,133] met all five procedure-specific quality-reporting criteria, of which 6 of 20 articles were published in a surgical journal and 8 of 20 articles in an oncology journal. Most articles (N = 222, 65%) met one or no procedure-specific quality-reporting criteria.

One hundred fifty-nine articles (47%) reported one or more complications or at least indicated the duration of follow-up for the assessment of complications. One hundred eighty-one (51%) of the articles reported morbidity and mortality rates including the complications leading to death. The death rates occurred most frequently in reports of postoperative morbidity and mortality (19 articles) and least frequently in descriptions of the predictive value of different proteins (0 articles) or radiological features (0 articles). Morbidity risk factor stratification based on factors such as the American Society of Anesthesia classification, body mass index, age, and comorbid diseases for characterising the patient population was described in 128 articles (38%).

Specific reporting criteria and reporting compliance according to the study’s primary endpoint are tabulated in Table 3. Of the articles with the primary endpoint of OS (N = 73), 42 articles (58%) met less than half (less than 7) of the 14 reporting criteria (median 7 criteria, IQR 4). Procedure-specific quality was reported in only 32 of these 73 articles (44%), with the macroscopic integrity of the mesorectal excision plane being the worst rated (N = 8, 11%). The majority of articles reporting disease-free survival (DFS) (30 of 51 articles, 59%) met less than seven reporting criteria (mean = 5.9 ± 2.8 (SD); median = 6 (range 2–14)), including a failure to report the macroscopic integrity of mesorectal excision (44 of 51 articles, 86%), the distance between the tumour and CRM (42 of 51 articles, 82%), CRM status (39 of 51 articles, 76%), number of retrieved lymph nodes (40 of 51 articles, 78%), and surgical radicality according to the site of the primary tumour (30 of 51 articles, 59%). Only one study reporting the locoregional recurrence rate (LRR) met all 14 quality-reporting [122] criteria. The majority of LRR articles (49 of 73 articles, 67%) met seven or fewer reporting criteria (mean = 6.3 ± 3 (SD); median = 5.5 (range 2–14)).

Table 4 shows, for each time period and overall, the specific criteria and the fulfilment of the reporting requirements over time. Among the 73 articles published from 1984 to 2001, a median of 5 (range, 2–9) criteria were met; among the 73 articles published from 2002 to 2009, a median of 6 (range, 2–11) criteria were satisfied; and among the 194 articles published from 2010 to 2022, a median of 6 (range, 2–14) criteria were met. The reporting quality pertaining to the procedure-specific criteria has shown little to no improvement over the years. During the intervals 2000–2004, 2005–2009, 2010–2014, 2015–2019, and 2020–2022, the median criteria values were 0 (range 0–4), 1 (range 0–4), 1 (range 0–5), 1 (range 0–5), and 1 (range 0–5), respectively (Appendix A).

The median value of the criteria met for RCTs published in oncology journals was 6 (range 2–14), and it was 7 (range 2–14) for those in surgical journals, 4.5 (range 2–11) for those in medical journals, and 5 (range 0–13) for those in radiology journals (Table 5).

## 4. Discussion

In this critical review of the multimodal rectal cancer treatment literature from over 40 years, we demonstrate considerable inconsistency in the quality of reporting of surgical outcome and complication data. The quality of surgical outcomes and reporting of complication data in articles on RCTs on rectal cancer are essential in the assessment of outcomes and areas for improving the quality of surgical care. An accurate appraisal of surgical quality requires consistency and reliability in the reporting of outcomes, data on specific complications, and procedure-specific quality.

In the study presented here, only 44% of the RCTs related to multimodal rectal cancer treatment reported on a minimum of 7 out of 14 critical surgical reporting criteria. Further, only half of the articles (N = 168, 49%) met the procedure-specific quality criteria reported in the article. Interestingly, surgical articles did not have a higher rate of better compliance regarding reporting procedure-specific quality criteria compared to oncology journals. Only 53% of the articles adequately defined the complications reported in the study. There was substantial variability in the published articles regarding what constituted a complication. The reporting of the complication severity using grading criteria was very limited (15%), and appropriate consideration of risk factors for surgical complications was described in only 38% of the articles. This lack of surgical quality reporting poses a significant challenge to synthesising and analysing the data for surgical interventions and multimodal treatment and potentially confounds the comparison of outcomes across studies.

Consistency in surgical quality criteria reporting would facilitate data comparison across studies and enable authors to perform better meta-analyses.

Although significant progress has been made in improving the reporting of surgical outcomes, our study indicates that further efforts are needed. Our results support the work of others [13,134] who have demonstrated the limitations in how surgical quality outcomes and complications are reported in the surgical literature.

The current study emphasises the need for the use of standard grading systems [11,12] to characterise the severity of complications after rectal cancer surgery. In the surgical literature, there are two areas that have received substantial attention regarding complication reporting: measuring the severity of complications and developing standardised outcome measures and clear definitions of complications. For example, the term “surgical site infection” provides a broad spectrum of different diagnoses. Data on “surgical site infection” are, therefore, only comparable between different studies or further computed in meta-analyses with caution.

The consistent reporting of standardised procedure-specific quality criteria is of equal importance to reporting the severity of complications. Curative surgical therapy of rectal cancer usually requires partial or complete excision of the mesorectum and, thus, the regional lymphatic drainage area and resection of the primary tumour in healthy tissue. Surgical therapy should include the following principles: complete mesorectal excision, CRM status, R0 status depending on the location of the primary tumour, a distance between the tumour and CRM of >1 mm, sufficient lymph nodes removed, and complete resection of the mesorectum. All these procedure-specific quality criteria influence OS, DFS, and LLR. However, our study shows that current strategies for reporting these data complicate the comparison of different multimodal therapy regimens and, thus, could be better. For example, if an attempt is made to compare two neoadjuvant therapy protocols wherein the surgical quality is described in one study but not in the other, it is difficult to determine which effect is from the neoadjuvant therapy and which is from the surgery.

When evaluating the procedure-specific quality criteria, it is important to note that the treatment of rectal cancer has changed dramatically over the past 40 years; nevertheless, it must be assumed that years or even decades are required before a novel surgical technique becomes an accepted standard. TME was introduced in 1982 [2], and the distance between the tumour and circumferential resection margin was introduced in 1986 [135]. It took several years to gain acceptance, so TME has only been considered the procedure of choice since 2004 [136]. It can be assumed that studies published before 2004 describe many fewer procedure-specific quality criteria than studies published after 2004 because the criteria did not yet exist. Interestingly, we found that in studies published after 2004 and even after 2020, the rate of reporting procedure-specific quality criteria remained low, at 15–45%, depending on the criterion.

The limitation of the study was that only randomized control trials were used. Selection bias may limit the generalizability of our findings. More recent articles may reflect improvements in complication reporting not captured in this study. A further limitation is the evaluation criteria used. The criteria used here were previously published and utilised in several studies to evaluate complication data. However, we acknowledge that these criteria are not universally accepted. Unfortunately, while checklists such as the STROBE statement [20] exist for evaluating what items should be reported in different studies, there are no universally accepted guidelines for assessing the quality of reporting on specific complications. Thus, we chose to use these criteria and modify them because they represent a practical and easy-to-understand strategy for characterising the quality of complication reporting. We further acknowledge that some criteria are not ubiquitous in their relevance. For example, it is not always possible to account for complication risk factors if the relationship to the outcome is not established. Another possible limitation is the time period that was chosen. The introduction of three procedure-specific quality criteria after 1982 may have had an effect on the results. To counteract this, we performed a subgroup analysis. The time periods were as follows: 1984–2001 and 2002–2022. The year 2001 was chosen as the reference period because this was the time when these procedures were performed—specific criteria already existed or were recommended in many guidelines. We also wanted to analyse how a surgical quality criterion became established in RCTs over time, so we chose the time when a method was first described.

This study demonstrates inconsistent quality in the reporting of surgical outcome and complication data in RCTs of multimodal rectal cancer treatment. The variability in reporting these data complicates the comparison of surgical outcomes between trials and potentially dilutes its importance for the further clinical development of the treatment.

## 5. Conclusions

The future development of guidelines for the accurate, comparable, and reproducible collection and reporting of quality surgical data is essential for future RCTs on multimodal rectal cancer treatment.

## Figures and Tables

**Figure 1 cancers-16-00026-f001:**
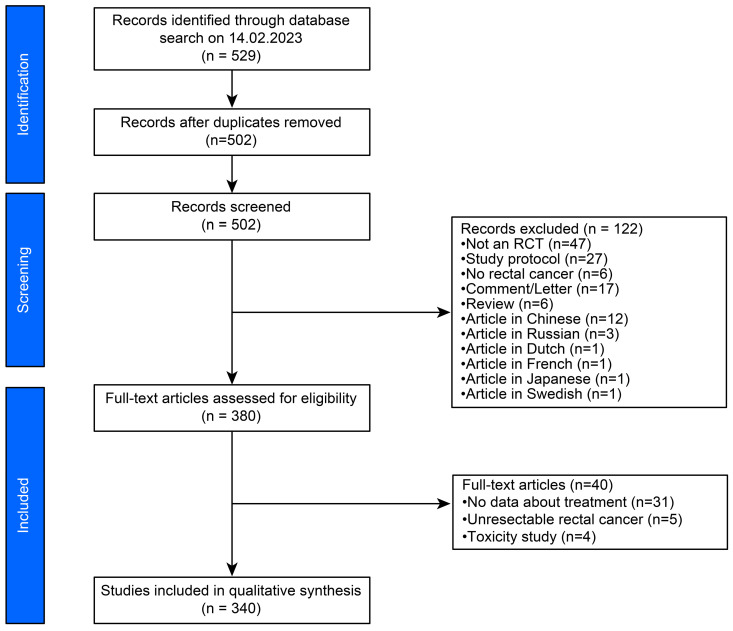
PRISMA flowchart of the literature research.

**Figure 2 cancers-16-00026-f002:**
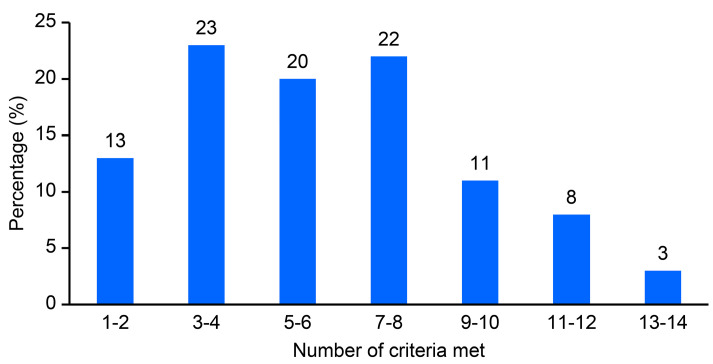
Completeness of surgical outcomes and complication data reporting in a convenience sample of the rectal treatment literature.

**Table 1 cancers-16-00026-t001:** Criteria utilised to evaluate surgical quality reporting in randomised clinical trials of the treatment of rectal cancer (modified from Martin RCG) [13].

Requirement	Points
Study design/method of accruing data	
Article indicates whether the data were collected prospectively or retrospectively.	+1
Article does not specify whether the data were collected prospectively or retrospectively.	0
Duration of follow-up	
Article indicates how long the patients were followed and evaluated for complications.	+1
Article does not specify how long the patients were observed and evaluated for complications.	0
Definition of complications	
Article defines at least one complication with specific inclusion criteria.	+1
Article does not define the complications.	0
Mortality rate and causes of death	
Number of patients who died in the postoperative period of study are recorded together with cause of death.	+1
Mortality data are not provided.	0
Morbidity rate and total complications	
Number of patients with any complication and the total number of complications are recorded.	+1
Morbidity data are not provided.	0
Severity grade utilized	
Any grading system designed to clarify the severity of complications including “major and minor” is reported.	+1
No arbitrary grading system is given to clarify the severity of complications including “serious and minor.”	0
Procedure-specific quality	
Circumferential resection margin- (CRM-) status	
Circumferential resection margin- (CRM-) status described.	+1
Circumferential resection margin- (CRM-) status not described.	0
Surgical radicality according to the site of the primary tumour ^a^	
Surgical radicality according to the site of the primary tumour described.	+1
Surgical radicality according to the site of the primary tumour not described.	0
Distance between the tumour and circumferential resection margin	
CRM distance described.	+1
CRM distance not described.	0
Number of retrieved lymph nodes	
Number of retrieved lymph nodes described.	+1
Number of retrieved lymph nodes not described.	0
Macroscopic intactness of mesorectal excision ^b^	
Macroscopic intactness of mesorectal excision described.	+1
Macroscopic intactness of mesorectal excision not described.	0
Type of surgery	
Type of surgery described.	+1
Type of surgery not described.	0
Length of stay	
Median or mean length of stay indicated in the study.	+1
No data on mean or average length of stay.	0
Risk factors included in the analysis	
Evidence of risk stratification and method used indicated in the study.	+1
No information on risk stratification or the method used is given in the study.	0

Notes: ^a^ R0 vs. R1 + R2 vs. unresected patients. ^b^ The non-peritonealized surface of the fresh specimen is examined circumferentially, and the completeness of the mesorectum is scored as follows: complete vs. near complete vs. incomplete vs. cannot be determined. Adapted from Martin et al. [13].

**Table 2 cancers-16-00026-t002:** Number of articles for each Jadad scale item [10].

Jadad Scale Items	Oncology Journal, n (%) n = 135	Surgical Journal, n (%) n = 69	Medical Journal, n (%) n = 23	Radiology Journal, n (%) n = 69	Gastroenterology Journal, n (%) n = 41	Total, n (%) n = 340
Randomisation						
Study described as randomised	135 (100)	69 (100)	23 (100)	69 (100)	41 (100)	340 (100)
Randomisation method described and appropriate	135 (100)	69 (100)	23 (100)	69 (100)	41 (100)	340 (100)
Randomisation method described and inappropriate	0 (0)	0 (0)	0 (0)	0 (0)	0 (0)	0 (0)
Randomisation method not described	0 (0)	0 (0)	0 (0)	0 (0)	0 (0)	0 (0)
Blinding						
Study described as double-blind (or single-blind)	0 (0)	1 (1.4)	1 (4.3)	2 (2.9)	1 (2.4)	5 (1.5)
Blinding method described and appropriate	135 (100)	68 (98.6)	22 (95.7)	67 (97.1)	40 (97.6)	335 (98.5)
Blinding method described and inappropriate	0 (0)	0 (0)	0 (0)	0 (0)	0 (0)	0 (0)
Blinding method not described	0 (0)	0 (0)	0 (0)	0 (0)	0 (0)	0 (0)
Study not described as blind	0 (0)	0 (0)	0 (0)	0 (0)	0 (0)	0 (0)
Withdrawals and dropouts						
Withdrawals and dropouts described	135 (100)	69 (100)	23 (100)	67 (100)	41 (100)	340 (100)
Withdrawals and dropouts not described	0 (0)	0 (0)	0 (0)	0 (0)	0 (0)	0 (0)

Notes: Adapted from Jadad et al. [10].

**Table 3 cancers-16-00026-t003:** Specific criteria and reporting compliance according to the primary endpoint.

Reporting Criterion	Primary Endpoint												Total, n (%)
	Overall Survival, n (%) n = 73	Disease-Free Survival, n (%) n = 51	Locoregional Recurrence, n (%) n = 73	Pathological Complete Response, n (%) n = 51	Pathological Features, n (%) n = 12	R0-Resection Rate, n (%) n = 8	Treatment-Related Toxicity, n (%) n = 24	Postoperative Morbidity and Mortality, n (%) n = 19	Functional Outcome, n (%) n = 19	Quality of Life, n (%) n = 18	Predictive Value of Different Proteins, n (%) n = 17	Radiological Features, n (%) N = 4	N = 340
Method of accruing data defined	73 (100)	51 (100)	73 (100)	51 (100)	12 (100)	8 (100)	24 (100)	19 (100)	19 (100)	18 (100)	17 (100)	4 (100)	340 (100)
Duration of follow-up indicated	73 (100)	51 (100)	73 (100)	51 (100)	12 (100)	8 (100)	24 (100)	19 (100)	19 (100)	18 (100)	17 (100)	4 (100)	340 (100)
Definitions of complications provided	35 (48)	24 (47)	33 (45)	29 (57)	2 (17)	7 (88)	10 (42)	19 (100)	11 (58)	5 (28)	0 (0)	0 (0)	159 (47)
Mortality rate, cause of death listed	39 (53)	25 (49)	39 (53)	32 (63)	2 (17)	7 (88)	11 (46)	19 (100)	10 (53)	6 (33)	0 (0)	0 (0)	174 (51)
Morbidity rate and total complications reported	37 (51)	24 (47)	38 (52)	31 (61)	2 (17)	7 (88)	10 (42)	19 (100)	12 (63)	6 (33)	1 (6)	0 (0)	173 (51)
Severity grade utilized	8 (11)	9 (18)	5 (7)	13 (25)	2 (17)	3 (38)	4 (17)	4 (21)	3 (16)	3 (17)	0 (0)	0 (0)	51 (15)
Procedure-specific quality													
CRM-status	22 (30)	12 (24)	35 (48)	31 (61)	9 (75)	7 (88)	4 (17)	4 (21)	3 (16)	4 (22)	5 (29)	1 (25)	128 (38)
Surgical radicality according to the site of the primary tumour ^a^	19 (26)	21 (41)	17 (23)	26 (51)	1 (8)	7 (88)	2 (8)	6 (32)	3 (16)	2 (11)	5 (29)	0 (0)	100 (29)
Distance between the tumour and CRM	16 (22)	9 (18)	21 (29)	22 (43)	6 (50)	5 (63)	1 (4)	5 (26)	2 (11)	4 (22)	5 (29)	1 (25)	90 (26)
Number of retrieved lymph nodes	15 (21)	11 (22)	18 (25)	17 (33)	7 (58)	2 (25)	0 (0)	4 (21)	2 (11)	1 (6)	0 (0)	0 (0)	70 (21)
Macroscopic intactness of mesorectal excision ^b^	8 (11)	7 (14)	12 (16)	10 (20)	8 (67)	2 (25)	2 (8)	3 (16)	0 (0)	1 (6)	6 (35)	1 (25)	57 (17)
Type of surgery indicated	44 (60)	33 (65)	56 (77)	36 (71)	10 (83)	5 (63)	13 (54)	17 (89)	11 (58)	12 (67)	4 (24)	1 (25)	222 (65)
Length of stay data reported	11 (15)	8 (16)	11 (15)	9 (18)	2 (17)	3 (38)	3 (13)	13 (68)	2 (11)	1 (6)	0 (0)	0 (0)	61 (18)
Risk factors included in analysis	17 (23)	17 (33)	29 (40)	24 (47)	3 (25)	8 (100)	7 (29)	15 (79)	9 (47)	6 (33)	2 (12)	1 (25)	130 (38)

Abbreviations: DFS: disease-free survival; LRR: locoregional recurrence rate; MoMo: postoperative morbidity and mortality; OS: overall survival; pCR: pathological complete response; CRM: Circumferential resection margin. Notes: ^a^ R0 vs. R1 + R2 vs. unresected patients. ^b^ The non-peritonealized surface of the fresh specimen is examined circumferentially, and the completeness of the mesorectum is scored as follows: complete vs. near complete vs. incomplete vs. cannot be determined. Adapted from Martin et al. [13].

**Table 4 cancers-16-00026-t004:** Specific criteria and reporting compliance according to time.

Reporting Criteria	Time Period Study Published								Total
	1984–1989	1990–1994	1995–1999	2000–2004	2005–2009	2010–2014	2015–2019	2020–2022	
	N = 11	N = 19	N = 28	N = 33	N = 55	N = 71	N = 83	N = 40	N = 340
Method of accruing data defined	11 (100%)	19 (100%)	28 (100%)	33 (100%)	55 (100%)	71 (100%)	83 (100%)	40 (100%)	340 (100%)
Duration of follow-up indicated	11 (100%)	19 (100%)	28 (100%)	33 (100%)	55 (100%)	71 (100%)	83 (100%)	40 (100%)	340 (100%)
Definitions of complications provided	7 (64%)	10 (53%)	13 (46%)	11 (33%)	26 (47%)	33 (47%)	35 (42%)	24 (60%)	159 ((47%)
Mortality rate, cause of death listed	8 (73%)	15 (79%)	16 (57%)	13 (39%)	26 (47%)	33 (47%)	37 (45%)	25 (63%)	174 (51%)
Morbidity rate and total complications	8 (73%)	12 (63%)	15 (54%)	12 (36%)	29 (53%)	33 (47%)	39 (47%)	25 (63%)	173 (51%)
Severity grade utilized	0 (0%)	1 (5%)	1 (4%)	3 (9%)	6 (11%)	6 (8%)	18 (22%)	16 (40%)	51 (15%)
Procedure-specific quality									
CRM status	0 (0%)	3 (16%)	7 (25%)	14 (42%)	20 (36%)	33 (47%)	33 (40%)	18 (45%)	128 (38%)
Surgical radicality according to site of primary tumour ^a^	0 (0%)	0 (0%)	1 (4%)	4 (12%)	11 (20%)	34 (48%)	36 (43%)	14 (35%)	100 (29%)
Distance between tumour and CRM	0 (0%)	0 (0%)	1 (4%)	5 (15%)	12 (22%)	28 (39%)	29 (35%)	15 (38%)	90 (26%)
Numbers of retrieved lymph nodes	0 (0%)	0 (0%)	3 (11%)	5 (15%)	17 (31%)	21 (30%)	18 (22%)	6 (15%)	70 (21%)
Macroscopic intactness of mesorectal excision ^b^	0 (0%)	0 (0%)	1 (4%)	5 (15%)	3 (5%)	12 (17%)	24 (29%)	12 (30%)	57 (17%)
Type of Surgery	9 (82%)	16 (84%)	17 (61%)	24 (73%)	34 (62%)	49 (69%)	47 (57%)	26 (65%)	222 (65%)
Length of stay data reported	7 (64%)	4 (21%)	1 (4%)	3 (9%)	7 (13%)	11 (17%)	19 (23%)	9 (23%)	61 (18%)
Risk factors included in analysis	1 (9%)	4 (21%)	5 (18%)	5 (15%)	18 (33%)	23 (32%)	42 (51%)	32 (80%)	130 (38%)

Abbreviations: CRM: Circumferential resection margin. Notes: ^a^ R0 vs. R1 + R2 vs. unresected patients; ^b^ The non-peritonealized surface of the fresh specimen is examined circumferentially, and the completeness of the mesorectum is scored as follows: complete vs. near complete vs. incomplete vs. cannot be determined. Adapted from Martin et al. [13].

**Table 5 cancers-16-00026-t005:** Specific criteria and reporting compliance according to the type of journal.

Reporting Criterion	Type of Journal						Total, n (%)
	Oncology, n (%) n = 135	Surgical, n (%) n = 69	Medical, n (%) n = 23	Radiology, n (%) n = 69	Gastroenterology, n (%) n = 41	Other, n (%) n = 3	n = 340
Method of accruing data defined	135 (100)	69 (100)	23 (100)	69 (100)	41 (100)	3 (100)	340 (100)
Duration of follow-up indicated	135 (100)	69 (100)	23 (100)	69 (100)	41 (100)	3 (100)	340 (100)
Definitions of complications provided	57 (42)	45 (65)	7 (30)	22 (32)	27 (66)	1 (33)	159 (47)
Mortality rate, cause of death listed	63 (47)	44 (64)	11 (48)	25 (36)	30 (73)	1 (33)	174 (51)
Morbidity rate and total complications reported	61 (45)	45 (65)	10 (43)	28 (41)	28 (68)	1 (33)	173 (51)
Severity grade utilized	17 (13)	11 (16)	1 (4)	10 (14)	12 (29)	0 (0)	51 (15)
Procedure-specific quality							
CRM status	54 (40)	26 (38)	5 (22)	25 (36)	17 (41)	1 (33)	128 (38)
Surgical radicality according to the site of the primary tumour ^a^	51 (38)	14 (20)	3 (13)	20 (29)	12 (29)	0 (0)	100 (29)
Distance between the tumour and CRM	38 (28)	21 (30)	3 (13)	16 (23)	12 (29)	0 (0)	90 (26)
Number of retrieved lymph nodes	25 (19)	13 (19)	5 (22)	15 (22)	11 (27)	1 (33)	70 (21)
Macroscopic intactness of mesorectal excision ^b^	25 (19)	13 (19)	2 (9)	7 (10)	9 (22)	1 (33)	57 (17)
Type of surgery	84 (62)	54 (78)	13 (57)	40 (58)	31 (76)	0 (0)	222 (65)
Length of stay data reported	23 (17)	22 (32)	3 (13)	3 (4)	10 (24)	0 (0)	61 (18)
Risk factors included in analysis	48 (36)	32 (46)	7 (30)	17 (25)	26 (63)	0 (0)	130 (38)

Abbreviations: DFS: disease-free survival; LRR: locoregional recurrence rate; MoMo: postoperative morbidity and mortality; OS: overall survival; pCR: pathological complete response; CRM: Circumferential resection margin. Notes: ^a^ R0 vs. R1 + R2 vs. unresected patients. ^b^ The non-peritonealized surface of the fresh specimen is examined circumferentially, and the completeness of the mesorectum is scored as follows: complete vs. near complete vs. incomplete vs. cannot be determined. Adapted from Martin et al. [13].

## Data Availability

The authors confirm that the data supporting the findings of this study are available within the article and its Appendix A.

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
