# Peer review of "Quality of Surgical Outcome Reporting in Randomised Clinical Trials of Multimodal Rectal Cancer Treatment: A Systematic Review"

_cancers, 2023, doi:10.3390/cancers16010026_

Round 1
Reviewer 1 Report
Comments and Suggestions for Authors
The aim of this paper was to analyse systematically the completeness rate of reporting of surgical outcomes and complication data in randomised controlled trials (RCTs) on rectal cancer. For a start the time intervall from 1982 to 2022 was chosen, finally 340 studies from 1984 to 2022 could be included for evaluation. The authors concluded that inconsistent surgical outcome and complication data reporting in multimodal rectal cancer treatment studies is standard. Only 2% (or 7/340) investigated RCTs met all 14 reporting criteria which were regarded as extremely important.
The main point of criticism is that the authors included parameters for all analysed RCTs from 1984 to 2022 irrespective of their introduction in studies or in pathological documentation. The CRM classification and the TME specimen quality classification were only established area-wide in the first decade of the new millenium (this refers to 3 of 14 criteria points, table 1), therefore this should not be judged negatively for RCTs before this time. The overall standards never can be raised for an earlier phase.
Specific comments:
Why did the authors start their seach already in 1982? As you wrote in line 95 Heald demonstrated his results in 1982 for the first time. In 1988 the term „holy plane“ was coined by him in another publication. As written in line 330 the TME concept realistically was implemented world-wide foremost in the new millennium. In the outgoing 20th century rectal cancer surgery area-wide neither was standardized nor always part of a multimodal concept of therapy.
Quirke demonstrated oncologic benefits of a good TME specimen. The Dutch TME trial (26, Kapiteijn et al.), recruited 1996-1999, published in 2001, was the first RCT which investigated the TME quality. In contrast, the German CAO/ARO/AIO-94 trial (27, Sauer et al.), published in 2004, did not take this into account. Realistically the importance of the TME quality was perceived not earlier than in the first decade. The same applies to the „CRM“-classification which area-wide was used routinely only in the new millenium.
Line 328: the authors write that „CRM“ was introduced in 1986. This cited paper of Quirke relates to „LRM“, the positivity of the lateral resection margin. Doesn´t that mean the R status (R0/R1)?
How did the authors chose the compared time intervals (lines 264-270): 1984-1997, 1998-2009, 2009-2022 (overlapping 2009)? As TME specimen quality and CRM status raised the awareness at first in the noughties wouldn´t it be better to compare the time periods e.g. from 1984-2002 and 2003-2022 (or in a similar way)?
The heading of table 4 reads "Specific criteria and reporting compliance according to time and type of journal". Unfortunately time intervals are not provided.
The authors included 340 studies for analysis, the list of references only mentions 140 citations. Shouldn´t the selected 340 papers be available at least in a supplementary?
Author Response
1 |
Reviewer's comment |
The aim of this paper was to analyse systematically the completeness rate of reporting of surgical outcomes and complication data in randomised controlled trials (RCTs) on rectal cancer. For a start the time intervall from 1982 to 2022 was chosen, finally 340 studies from 1984 to 2022 could be included for evaluation. The authors concluded that inconsistent surgical outcome and complication data reporting in multimodal rectal cancer treatment studies is standard. Only 2% (or 7/340) investigated RCTs met all 14 reporting criteria which were regarded as extremely important. The main point of criticism is that the authors included parameters for all analysed RCTs from 1984 to 2022 irrespective of their introduction in studies or in pathological documentation. The CRM classification and the TME specimen quality classification were only established area-wide in the first decade of the new millenium (this refers to 3 of 14 criteria points, table 1), therefore this should not be judged negatively for RCTs before this time. The overall standards never can be raised for an earlier phase.
|
|
Our comment
|
We appreciate the reviewer's comments and criticisms, which require further clarification. We fully agree with the reviewer that it takes time for new surgical quality criteria to become generally accepted as the gold standard, but no data are currently available on how long it takes for a procedure to become generally standardized. The analysis of how a criterion is described in RCTs over time from initial publication was a secondary outcome of our study. In order to illustrate this, we have included this point in the Methods section and have created an additional table in the Results section that shows the percentage of RCTs that each parameter represents in relation to time. We are also in agreement with the reviewer that 3 of the 14 criteria items were only introduced after 1982. Therefore, RCTs prior to this date should not be judged negatively. We already had a discussion of this issue in the discussion section, but not in sufficient detail. We performed a new analysis to address this important point. As recommended by the reviewer in comment 5, we divided the patients into two time periods. The time periods were as follows: 1984-2001 and 2002-2022. 2001 was chosen as the cut-off point because this was the time when these procedures were being performed - specific criteria were already in place or were recommended in many guidelines.
|
|
Alterations in manuscript
|
Please see the changes on pages 4 lines 134-136 in the methods section of the amended manuscript:
· (…) An additional outcome measure was to objectively evaluate changes in procedure specific quality criteria over time, particularly with respect to their initial consultation in previous literature. (…)
Please see the changes on pages 10 lines 275-284 in the results section of the amended manuscript:
· (…) Table 4 shows, for each time period and overall, the specific criteria and the fulfillment of reporting requirements over time. Among the 73 49 articles published from 1984 to 2001 1997, a median of 5 (range, 2-9) criteria were met; among the 73 97 articles published from 1998 2002 to 2009, a median of 6 (range, 2-11) criteria were satisfied; and among the 194 articles published from 2010 2009 to 2022, a median of 6 (range, 2-14) criteria were met. The reporting quality pertaining to the procedure-specific criteria has shown little to no improvement over the years. During the intervals 2000-2004, 2005-2009, 2010-2014, 2015-2019, and 2020-2022, the median criteria values were 0 (range 0-4), 1 (range 0-4), 1 (range 0-5), 1 (range 0-5), and 1 (range 0-5), respectively (Supplementary Figure 2)
Please see the newly inserted Table 4 on page 10 lines 286-289 in the results section of the amended manuscript:
· Table 4. Specific criteria and reporting compliance according to time
Please see the newly inserted Supplementary Figure 2 on page 10 line 283 -284 in the results section of the amended manuscript:
· Supplementary Figure 2: Analysis of Procedure-specific quality criteria met from 1984 to 2022. The figure demonstrates no significant linear trend between the quality of articles.
Please see the changes on pages 13 lines 375-282 in the discusion section of the amended manuscript:
· Another possible limitation is the time period that was chosen. The introduction of three procedure-specific quality criteria after 1982 may have had an effect on the results. To counteract this, we performed a subgroup analysis. The time periods were as follows: 1984-2001 and 2002-2022. 2001 was chosen as the reference period because this was the time when these procedures were performed - specific criteria already existed or were recommended in many guidelines. We also wanted to analyze how a surgical quality criterion became established in RCTs over time, so we chose the time when a method was first described.
|
2 |
Reviewer's comment |
Why did the authors start their seach already in 1982? As you wrote in line 95 Heald demonstrated his results in 1982 for the first time. In 1988 the term „holy plane“ was coined by him in another publication. As written in line 330 the TME concept realistically was implemented world-wide foremost in the new millennium. In the outgoing 20th century rectal cancer surgery area-wide neither was standardized nor always part of a multimodal concept of therapy.
|
|
Our comment
|
We thank the reviewer for this important question, which requires a detailed explanation. The introduction of CRM is a significant prognostic factor, as mentioned above. Our hypothesis was that the procedure-specific quality criterion in RCTs is high, so we aimed also to observe its historical behavior by selecting the point in time when it was first documented in the literature. An alternative would have been to select the time when a method was included in the guidelines. The problem is: 1. which guidelines (German vs. UK vs. US vs. Japanese, etc.) 2. a surgical method is not included in all guidelines simultaneously. Therefore, it has been difficult to come to a consensus. We have done our best to better explain the time interval from 1982 to 2022 in the methods section.
|
|
Alterations in manuscript
|
Please see the changes on pages 3 lines 96-101 in the methods section of the amended manuscript:
· (…) A subgroup analysis was performed to compare publications published before and after the year 2000, as the implementation of the TME concept and related procedure specific criteria occurred primarily on a global scale in the new millennium. Year 2001 was chosen as the cut-off point because this was the time when these procedures were being performed - specific criteria were already in place or were recommended in many guidelines.
|
3 |
Reviewer's comment |
Quirke demonstrated oncologic benefits of a good TME specimen. The Dutch TME trial (26, Kapiteijn et al.), recruited 1996-1999, published in 2001, was the first RCT which investigated the TME quality. In contrast, the German CAO/ARO/AIO-94 trial (27, Sauer et al.), published in 2004, did not take this into account. Realistically the importance of the TME quality was perceived not earlier than in the first decade. The same applies to the „CRM“-classification which area-wide was used routinely only in the new millenium.
|
|
Our comment
|
|
|
Alterations in manuscript
|
Please see the changes on pages 10 lines 275-284 in the results section of the amended manuscript:
· (…) Table 4 shows, for each time period and overall, the specific criteria and the fulfillment of reporting requirements over time. Among the 73 49 articles published from 1984 to 2001 1997, a median of 5 (range, 2-9) criteria were met; among the 73 97 articles published from 1998 2002 to 2009, a median of 6 (range, 2-11) criteria were satisfied; and among the 194 articles published from 2010 2009 to 2022, a median of 6 (range, 2-14) criteria were met. The reporting quality pertaining to the procedure-specific criteria has shown little to no improvement over the years. During the intervals 2000-2004, 2005-2009, 2010-2014, 2015-2019, and 2020-2022, the median criteria values were 0 (range 0-4), 1 (range 0-4), 1 (range 0-5), 1 (range 0-5), and 1 (range 0-5), respectively (Supplementary Figure 2)
Please see the newly inserted Table 4 on page 10 lines 286-289 in the results section of the amended manuscript:
· Table 4. Specific criteria and reporting compliance according to time
Please see the newly inserted Supplementary Figure 2 on page 10 line 283 -284 in the results section of the amended manuscript:
· Supplementary Figure 2: Analysis of Procedure-specific quality criteria met from 1984 to 2022. The figure demonstrates no significant linear trend between the quality of articles.
Please see the changes on pages 13 lines 375-282 in the discusion section of the amended manuscript:
· Another possible limitation is the time period that was chosen. The introduction of three procedure-specific quality criteria after 1982 may have had an effect on the results. To counteract this, we performed a subgroup analysis. The time periods were as follows: 1984-2001 and 2002-2022. 2001 was chosen as the reference period because this was the time when these procedures were performed - specific criteria already existed or were recommended in many guidelines. We also wanted to analyze how a surgical quality criterion became established in RCTs over time, so we chose the time when a method was first described.
|
4 |
Reviewer's comment |
Line 328: the authors write that „CRM“ was introduced in 1986. This cited paper of Quirke relates to „LRM“, the positivity of the lateral resection margin. Doesn´t that mean the R status (R0/R1)?
|
|
Our comment
|
We appreciate the point the Reviewer makes. To be more precise, in his paper published in The Lancet in 1986, Quirke was describing the distance between the tumor and the resection margin. The reviewer is also correct that R status is part of this.
|
|
Alterations in manuscript
|
Please see the changes on page 12 lines 354-356 in the Discusion section of the amended manuscript.
· (…)and surgical radicality according to the site of the primary tumour and the distance between the tumour and circumferential resection margin CRM was introduced in 1986
|
5 |
Reviewer's comment |
How did the authors chose the compared time intervals (lines 264-270): 1984-1997, 1998-2009, 2009-2022 (overlapping 2009)? As TME specimen quality and CRM status raised the awareness at first in the noughties wouldn´t it be better to compare the time periods e.g. from 1984-2002 and 2003-2022 (or in a similar way)?
|
|
Our comment
|
|
|
Alterations in manuscript
|
Please see the changes on pages 10 lines 275-280 in the results section of the amended manuscript:
· (…) Table 4 shows, for each time period and overall, the specific criteria and the fulfillment of reporting requirements over time. Among the 73 49 articles published from 1984 to 2001 1997, a median of 5 (range, 2-9) criteria were met; among the 73 97 articles published from 1998 2002 to 2009, a median of 6 (range, 2-11) criteria were satisfied; and among the 194 articles published from 2010 2009 to 2022, a median of 6 (range, 2-14) criteria were met.
Please see the newly inserted Table 4 on page 10 lines 286-289 in the results section of the amended manuscript:
· Table 4. Specific criteria and reporting compliance according to time
|
6 |
Reviewer's comment |
The heading of table 4 reads "Specific criteria and reporting compliance according to time and type of journal". Unfortunately time intervals are not provided.
|
|
Our comment
|
|
|
Alterations in manuscript
|
Please see the changes on page 10 line 286 and page 11 line 295 in the Results section of the amended manuscript:
· A new table 4 has been created.
· Table 4 has been replaced by table 5 New.
|
7 |
Reviewer's comment |
The authors included 340 studies for analysis, the list of references only mentions 140 citations. Shouldn´t the selected 340 papers be available at least in a supplementary?
|
|
Our comment
|
We thank the reviewer for this hint. As recommended by the reviewers, we have summarized all included studies in a table (Supplamantary Table S3).
|
|
Alterations in manuscript
|
Please see the changes on pages 6 line 190 in the Results section of the amended manuscript:
· The studies that have been included are summarized in Supplementary Table S3. |

Reviewer 2 Report
Comments and Suggestions for Authors
Overall, the article was a metanalysis of prior randomize control trials for surgical outcomes and complications of rectal cancer management, with a focused aim of the quality of these studies. Articles from the years 1982-2022 were found and out of those 340/509 studies showed qualitative data prompting further inclusion. Fourteen components were evaluated and points were awarded to studies that elicited these components in their research. Furthermore, the Jadad Scale was used to evaluate for randomization, blinding, and withdrawals/dropouts; points were also awarded to the articles that displayed these qualities. The authors went above and beyond to also evaluate which journals these articles came from and classified each based on their impact factor as of August 2023. In analysis, the authors assessed for commonly underreported criteria such as complication severity, macroscopic integrity of mesorectal excision, length of stay, lymph nodes retrieved, etc. Although the article aimed to achieve assessing the quality of each of these studies, it rather came across the lack of consistency with various studies evaluating their own extensive aims. In this case, only 44% of the trials reported at least 7/14 critical reporting data that authors suggested research should include. Although the data was not impressive in how research reliability isn’t the best, the authors of this paper did pursue an admirable subject of reliability of research which should be more consistent. They drew attention to a topic that doesn’t get enough attention, which is the use of standardized grading system to characterize journal articles. This article also highlighted the following principles to be included for rectal cancer surgical therapy: complete mesorectal excision, CRM status, R0 status depending on the location of the primary tumour, a distance between the tumour and CRM of > 1 mm, sufficient lymph nodes removed, and complete resection of the mesorectum. The limitation of the study was that only randomized control trials were used, which the paper did mention as a limitation. The article also mentions the limitation of the inclusion time period as research was gathered from 1982 to 2022, with negligence of articles prior to 1982 and after 2022. They also highlighted their criteria as a limitation as these aren’t universally accepted. Overall the authors did an excellent job in their metanalysis of a topic that is extremely difficult to study, with bringing a focus of how rectal cancer management is consistent and requires further research to pave a path for universally respected management and guidelines.
Author Response
1 |
Reviewer's comment |
Overall, the article was a metanalysis of prior randomize control trials for surgical outcomes and complications of rectal cancer management, with a focused aim of the quality of these studies. Articles from the years 1982-2022 were found and out of those 340/509 studies showed qualitative data prompting further inclusion. Fourteen components were evaluated and points were awarded to studies that elicited these components in their research. Furthermore, the Jadad Scale was used to evaluate for randomization, blinding, and withdrawals/dropouts; points were also awarded to the articles that displayed these qualities. The authors went above and beyond to also evaluate which journals these articles came from and classified each based on their impact factor as of August 2023. In analysis, the authors assessed for commonly underreported criteria such as complication severity, macroscopic integrity of mesorectal excision, length of stay, lymph nodes retrieved, etc. Although the article aimed to achieve assessing the quality of each of these studies, it rather came across the lack of consistency with various studies evaluating their own extensive aims. In this case, only 44% of the trials reported at least 7/14 critical reporting data that authors suggested research should include. Although the data was not impressive in how research reliability isn’t the best, the authors of this paper did pursue an admirable subject of reliability of research which should be more consistent. They drew attention to a topic that doesn’t get enough attention, which is the use of standardized grading system to characterize journal articles. This article also highlighted the following principles to be included for rectal cancer surgical therapy: complete mesorectal excision, CRM status, R0 status depending on the location of the primary tumour, a distance between the tumour and CRM of > 1 mm, sufficient lymph nodes removed, and complete resection of the mesorectum.
|
|
Our comment
|
We thank the reviewer for his favorable appraisal of our work.
|
|
Alterations in manuscript
|
Changes in the amended manuscript:
None
|
2 |
Reviewer's comment |
The limitation of the study was that only randomized control trials were used, which the paper did mention as a limitation
|
|
Our comment
|
We thank the reviewer for this hint. We also mentioned this important limitation, as suggested.
|
|
Alterations in manuscript
|
Please see the changes on pages 13 lines 364-365 in the discusion section of the amended manuscript:
· Our analysis was limited to RCTs. The limitation of the study was that only randomized control trials were used. Selection bias may limit the generalizability of our findings. More recent articles may reflect improvements in complication reporting not captured in this study.
|
3 |
Reviewer's comment |
The article also mentions the limitation of the inclusion time period as research was gathered from 1982 to 2022, with negligence of articles prior to 1982 and after 2022. They also highlighted their criteria as a limitation as these aren’t universally accepted. Overall the authors did an excellent job in their metanalysis of a topic that is extremely difficult to study, with bringing a focus of how rectal cancer management is consistent and requires further research to pave a path for universally respected management and guidelines.
|
|
Our comment
|
We thank the reviewer for his favorable appraisal of our work.
|
|
Alterations in manuscript
|
Changes in the amended manuscript:
None |

Round 2
Reviewer 1 Report
Comments and Suggestions for Authors
Dear authors,
all queries were elaborately answered. The manuscript has been well revised as well as the objected table. There are no more points of criticism on my part.